# A Global Perspective on Gastric Cancer Screening: Which Concepts Are Feasible, and When?

**DOI:** 10.3390/cancers15030664

**Published:** 2023-01-21

**Authors:** Wladyslaw Januszewicz, Maryla Helena Turkot, Peter Malfertheiner, Jaroslaw Regula

**Affiliations:** 1Department of Oncological Gastroenterology, Maria Sklodowska-Curie National Research Institute of Oncology, 02-781 Warsaw, Poland; 2Department of Gastroenterology, Hepatology and Clinical Oncology, Centre of Postgraduate Medical Education, 02-781 Warsaw, Poland; 3Department of Gastroenterology, Hepatology and Infectious Diseases, Otto-von-Guericke University, 39120 Magdeburg, Germany

**Keywords:** stomach neoplasms, early detection, helicobacter pylori

## Abstract

**Simple Summary:**

Although gradually declining in incidence, gastric cancer has remained a substantial disease burden and an epidemiological challenge worldwide. In this review article, we provide an update on existing screening programs in high-risk countries and put forth considerations for potentially applicable gastric cancer-screening options in intermediate- and low-risk regions.

**Abstract:**

Background: Gastric cancer (GC) remains the fifth most common cancer and the third most common cause of cancer-related death globally. In 2022, GC fell into the scope of the updated EU recommendations for targeted cancer screening. Given the growing awareness of the GC burden, we aimed to review the existing screening strategies for GC in high-risk regions and discuss potentially applicable modalities in countries with low-to-intermediate incidence. Methods: The references for this Review article were identified through searches of PubMed with the search terms “gastric cancer”, “stomach cancer”, “Helicobacter pylori”, and “screening” over the period from 1995 until August 2022. Results: As *Helicobacter pylori* (*H. pylori*)-induced gastritis is the primary step in the development of GC, the focus on GC prevention may be directed toward testing for and treating this infection. Such a strategy may be appealing in countries with low- and intermediate- GC incidence. Other biomarker-based approaches to identify at-risk individuals in such regions are being evaluated. Within high-incidence areas, both primary endoscopic screening and population-based *H. pylori* “test-and-treat” strategies represent cost-effective models. Conclusions: Given the significant variations in GC incidence and healthcare resources around the globe, screening strategies for GC should be adjusted to the actual conditions in each region. While several proven tools exist for accurate GC diagnosis, a universal modality for the screening of GC populations remains elusive.

## 1. Background

Gastric cancer (GC), despite its decreasing incidence over the last two decades, remains the third most common cause of cancer-related death worldwide [1]. Furthermore, GC is related to poor survival, with an age-standardized 5-year survival range of 20–40% [2], with the exceptions of South Korea and Japan, where it ranges between 60% to 70% [2]. This overall unfavorable prognosis is mainly related to an advanced stage of the disease at the time of presentation to physicians. On the contrary, early GC has an excellent prognosis (a 5-year survival rate of >90%) and can often be treated with minimally-invasive, organ-sparing modalities, such as endoscopic resection [3].

GC remains a highly heterogeneous disease from both morphological and molecular standpoints [4]. The recent WHO classification (2019) distinguishes several subtypes of gastric adenocarcinoma (e.g., papillary, tubular, mucinous, poorly cohesive carcinomas, etc.); however, these subtypes offer limited clinical applicability [5]. Furthermore, molecular profiling has led to the identification of four molecular subtypes of GC: (i) tumors positive for Epstein–Barr virus (EBV), (ii) a microsatellite-unstable (MSI) subtype, (iii) genomically stable (GS) tumors, and (iv) tumors with chromosomal instability (CIN) [6]. This tumor subtyping shows great potential in improving targeted cancer therapies and prediction models; however, thus far, its clinical utility remains to be further exploited.

In clinical practice, non-cardia GC (NCGC) remains the most common anatomical subtype of GC, which can be further subdivided into two histological categories—intestinal and diffuse—according to Lauren’s classification [7]. *Helicobacter pylori* (*H. pylori*) infection is the main trigger for GC, regardless of the histological phenotype. Other conditions—such as older age, high salt intake, a diet low in fruit and vegetables, smoking, and family history—contribute to gastric carcinogenesis if concomitant with *H. pylori* gastritis [8,9].

The development of intestinal-type NCGC is characterized by a stepwise progression described in the Correa cascade—an inflammatorily driven pathway typically initiated by the *H. pylori* infection. Then, the inflamed gastric mucosa undergoes several premalignant stages: chronic atrophic gastritis with intestinal metaplasia, dysplasia, and, eventually, intramucosal cancer [10]. This multistage process provides a unique opportunity for the screening and prevention of GC.

The diffuse type of NCGC follows a different oncogenic cascade, which is primarily genetically determined and less associated with environmental factors. Nevertheless, the development of these tumors may also be attributed to *H. pylori* infection, which makes this organism a common ground in gastric carcinogenesis [11,12,13]. Overall, *H. pylori* is responsible for ~90% of NCGC’s global burden [13,14] and is increasingly recognized as the primary target of GC prevention strategies. However, a series of individual considerations must be made to adopt and implement such a strategy.

## 2. The Rationale for Gastric Cancer Screening

A variety of factors must be considered when a population-wide cancer-screening program is set in place, the most important of which are the country’s demographic profile and the local burden of the disease of interest. In this context, the age-standardized incidence rate (ASR) is a commonly used parameter that characterizes the different risk levels in a given population [15]. In terms of GC, three risk areas can be determined (Figure 1) [16]:

High-risk areas: ASR ≥ 20 per 100,000 person-years (p-y), e.g., Japan, Korea, and China.

Intermediate risk areas: ASR ≥ 10 and <20 per 100,000 p-y, e.g., Portugal, Lithuania, Romania, and Slovenia.

Low-risk areas: ASR < 10 per 100,000 p-y, e.g., the USA, UK, Sweden, and Germany [16].

At present, population-wide screening for GC is performed only in high-risk regions. Although there is little evidence demonstrating that these programs have a beneficial effect on GC outcomes, a recent meta-analysis has indicated that endoscopic screening could lead to a 40% risk reduction in GC mortality in the high-risk Asian population (relative risk (RR) 0.60; 95% confidence intervals (CI): 0.49–0.73) [17]. The available data originate from countries with established national screening programs based on a primary endoscopic examination, namely, Japan, South Korea, and China (summarized in Table 1).

## 3. Primary Endoscopic Screening in High-Risk Areas

Japan began a nationwide GC-screening program in 1983, which was based on an upper gastrointestinal series with barium meal (UGIS). Initially, it was aimed at individuals ≥40 years with an annual examination frequency. For decades, radiography was the only recommended test for GC screening in Japan and the only modality with a documented impact on reducing GC mortality [23,24,25,26]. However, following the revisions of the Japanese Guidelines for Gastric Cancer Screening in 2014 and again in 2018, endoscopy was approved as an alternative primary screening method [18,19]. Additionally, the age threshold for screening’s initiation was raised to 50 years, and the screening intervals were stretched to 2–3 years [18]. Despite these changes, some modeling studies suggest that the Japanese program is still not cost-effective and instead should consider the triennial endoscopic screening of individuals aged 50 to 75 [27].

South Korea’s population-based screening program for GC was implemented in 1999 as part of the National Cancer-Screening Program (NCSP). The NCSP offers endoscopy or UGIS to citizens 40 years or older every two years (with no upper age limit) [20,28]. A recent report showed that the program resulted in an >20% reduction in GC mortality within the screened population (odds ratio [OR] 0.79; 95% CI: 0.77–0.81) [29], with most of the impact provided by the endoscopy screening alone (OR 0.53; 95% CI: 0.51–0.56), as opposed to the UGIS test, for which no benefit was shown (OR 0.98; 95% CI: 0.95–1.01) [29]. This finding may explain the growing preference for endoscopy over UGIS within the Korean screening program. The fraction of participants choosing endoscopy over UGIS has risen from 31.2% in 2002 to 72.6% in 2011 [22]. In South Korea (unlike Japan), endoscopy costs are comparable to photofluorography; hence, endoscopy remains the most cost-effective screening modality [28,30]. Overall, this country’s participation in the GC-screening program is steadily rising, with an estimated increase of 4.3% every year [22].

In China, the endoscopy-based screening program was launched in selected high-risk areas in 2008 [31]. The initial program targeted only those individuals between 40 and 69 years old who were residents of rural areas. However, in October 2012, the Chinese government initiated a population-based Cancer-Screening Program in Urban China (CanSPUC), which targeted five of the most prevalent types of cancers in metropolitan areas (lung, breast, liver, colorectal, and gastric cancers). The GC-screening method remained endoscopy-based, and the findings in the initial examination directed the follow-up recommendations for each individual. A recent cost-effectiveness analysis confirmed that endoscopic screening for upper gastrointestinal (UGI) cancers performed biannually may constitute a cost-effective modality in high-risk areas of China [32].

Lastly, it should be emphasized that immigrants from countries with high incidence of GC that migrate to low-incidence regions of GC retain their elevated cancer risk. For example, in the United States, two simulation studies based on a decision-analytic Markov model have shown that endoscopic screening for GC would remain cost-effective for Asian Americans aged 50 or older [33,34].

Although endoscopic GC screening has been well-established in some countries, significant challenges are associated with such a strategy. These are related to insufficient endoscopy services and budget constraints. Furthermore, this form of screening requires qualified endoscopists, and these professionals are often not available in sufficient numbers or are not adequately trained. Substantiating this, a recent meta-analysis showed that the pooled prevalence of missed upper GI cancers during endoscopy amounted to 10.7% (95% CI: 8.0–13.7%) [35].

Optimizing training in endoscopy and implementing artificial intelligence (AI) technology will undoubtedly contribute to reducing the number of missed lesions. AI platforms using deep-learning algorithms are increasingly being incorporated into endoscopy equipment, thus aiding in detecting and characterizing neoplastic lesions. A recent Chinese study used over a million endoscopic images from 84,424 individuals to develop and validate a dedicated AI system (Gastrointestinal Artificial Intelligence Diagnostic System–GRAIDS) in order to identify UGI neoplasia. Interestingly, the sensitivity of GRAIDS was as high as that of an expert endoscopist (sensitivity of 0.942 vs. 0.945; *p* = 0.692). Moreover, the support of GRAIDS increased the diagnostic accuracy of trainee endoscopists. This brings hope that such systems could help assist community-based hospitals with respect to improving their effectiveness in diagnosing upper gastrointestinal cancer [36].

With regard to GC, the utility of AI has been explored in several fields, including the endoscopic detection of *H. pylori* infection, chronic atrophic gastritis, early GC, cancer invasion depth prediction, and pathology recognition [37]. AI can also be used for procedural quality assessment. For example, providing real-time detection and feedback on the inspection process (e.g., identifying the blind spots missed during endoscopic evaluation or indicating low-quality images that require recapturing) may, in effect, aid in improving the quality and standardizing the procedure [38]. In summary, the concept of using AI in endoscopic screening appears very promising but requires future research and implementation trials before being introduced into an organized screening program [39].

## 4. Gastric Cancer Screening in Low- to Intermediate-Risk Countries

There is no rationale for wide-ranged endoscopic screening in countries with low GC incidence, and its benefit remains uncertain in countries with intermediate risk, thus raising the following question: should these countries offer a GC-screening program at all? If affirmative, another question follows: which kind of screening program should be implemented?

There have been several initiatives to uncover feasible screening strategies in countries with an intermediate risk for GC. In particular, a recent cost–utility analysis postulated that upper GI endoscopic screening may be cost-effective if combined with colonoscopy screening in individuals between 50 to 75 years [40]. While it is an interesting concept, most colorectal cancer (CRC)-screening programs worldwide are based on fecal occult blood tests (FOBTs) [41]. This would be highly limiting, as gastroscopic evaluation would be available only to those offered a colonoscopy after a positive FOBT. Moreover, as previously mentioned, it would only be cost-effective in countries where the incidence of GC is within the intermediate-risk range.

To date, most western countries suggest a tailored approach, in which endoscopy screening is restricted to only those with known risk factors for GC. For example, the British Society of Gastroenterology’s (BSG) guidelines consider endoscopy screening to be feasible for individuals aged >50 years with other high-risk features, namely, those of a male sex, who smoke, have pernicious anemia, and/or have a family history of GC [42]. Similarly, the recent Maastricht VI/Florence consensus suggests endoscopy with biopsies in asymptomatic individuals with a family history of GC at age ≥45 years [9].

Generally, the current role of endoscopy in the early detection of GC in the West represents that of a surveillance tool for pre-defined high-risk individuals rather than a screening modality to address the general population. As endoscopy is comparatively expensive and invasive; thus, a counterpart to aid in identifying these individuals with a cheaper, less invasive, and more patient-tolerable test is needed. Indeed, several concepts fulfilling these criteria are currently being debated.

*H. pylori* testing (and treatment).

*H. pylori* is a class I carcinogen according to the World Health Organization (WHO), and remains the primary risk factor for GC worldwide [43]. There are several non-invasive tests available in clinical routines for the detection of this pathogen, including the *H. pylori* stool antigen test (HPSA), the urea breath test (UBT), and serological tests (e.g., IgG to *H. pylori*). A population-based “test-and-treat” approach is easily accessible with these assays, rendering the strategy a method of choice for primary GC prevention in high-incidence areas [44]. Successful eradication therapy has been proven to cure gastric mucosal inflammation, thus preventing its progression to pre-neoplastic lesions [8]. Indeed, the latest recommendations from the Kyoto consensus for high-risk regions support active and early screening for *H. pylori* infection before chronic inflammation and its complications occur [45].

Furthermore, a recent report from Taiwan has shown that HPSA testing could be successfully coupled with a CRC-screening program [46]. In this nationwide study, eligible individuals were randomly invited to receive either the standard fecal immunochemical test (FIT) or an FIT supplemented with an HPSA test. A positive HPSA result was followed by eradication therapy, which was successful in 91.9% (95% CI: 91.1–92.7%) of patients in an intention-to-treat (ITT) analysis. Interestingly, the addition of the HPSA test increased the screening program’s participation rate by 13.9% (95% CI: 13.4–14.4%) compared to FIT-only testing. It is also worth noting that *H. pylori* carriers had a higher rate of colorectal adenomas than non-carriers (adjusted RR 1.15; 95% CI: 1.03–1.28, *P* = 0.01) [46]. Although the baseline results did not show any difference in the detection rates of early-stage GC between the groups, the long-term outcomes (e.g., the study’s effect on GC mortality) are still awaited. As most organized CRC-screening programs are based on FOBT (particularly FIT) testing [41], this approach could be easily adaptable in those countries. However, it must be emphasized that many countries still do not conduct population-based CRC screening or have a low screening uptake. Moreover, countries with opportunistic programs usually base their results on colonoscopy as an initial test [41].

Despite the many benefits of “test-and-treat” strategies, several concerns are inherent to such concepts. Most importantly, the increase in the consumption of antibiotics may result in a higher rate of antimicrobial resistance within *H. pylori*, which is already a documented issue. To illustrate this, a survey including 24 centers across Europe showed that the primary form of antimicrobial resistance of *H. pylori* to clarithromycin has doubled in the last two decades [47]. Clearly, any endeavor to “test-and-treat” *H. pylori* should consider an appropriate selection of eradication regimens with the highest possible efficacy and affordability.

### Serological Biomarkers

Both atrophic gastritis and intestinal metaplasia are associated with an increased risk of GC development. The prevalence and malignant potential of these conditions remain poorly characterized in western countries; however, a nationwide cohort study in the Netherlands has shown an annual progression rate of 0.1% and 0.25% for atrophic gastritis and intestinal metaplasia, respectively [48]. So far, endoscopy remains the mainstay for diagnosing and surveilling precancerous gastric conditions; in fact, European guidelines (ESGE) recommend endoscopic monitoring every three years for patients with extensive atrophy or intestinal metaplasia [49,50]. However, serological biomarkers may provide a promising alternative with respect to identifying precancerous gastric conditions.

Pepsinogen I and II (PGI and PGII), both precursors of pepsin, are produced by the gastric mucosa and released into the gastric lumen and peripheral circulation [51]. PGI is secreted mainly by the chief and mucous neck cells in the fundic glands, and PGII is secreted by the pylorus cells. When atrophic changes develop in the gastric corpus, the level of PGI decreases while PGII levels remain relatively stable or may even increase. Hence, a low serum pepsinogen level or PGI/PGII ratio is a specific indicator of extensive chronic atrophic gastritis.

Most studies on the utility of PG testing originate from high-risk areas for GC [52]. For example, although not recommended by the revised Japanese Guidelines [18] for GC-screening purposes, the “ABC method”—a combined serum assay for anti-*H. pylori* IgG antibody and PG levels—has successfully been used for GC risk assessment in Japanese patients [53].

While serological testing has not yet penetrated into broader clinical practice in low-risk countries (e.g., the BSG guidelines do not recommend using biomarkers as a screening tool within the UK [42]), several initiatives to implement serological testing for screening have been recently undertaken. For example, a report from the USA showed that non-invasive screening with serum PG may reduce GC mortality in high-risk individuals (actively smoking men aged >50 years) and remain cost-effective [54]. Another study from the US showed that PG(+) individuals had an overall 8.5-fold increased risk of GC and an 11-fold increased risk of NCGC specifically (OR, 11.1; 95% CI: 4.3–28.8) compared to PG(-) cases. In general, a positive PG status yielded low sensitivity but high specificity for both non-cardia (44.3%; 93.6%) and cardia gastric cancers (5.7%; 97.2%) [55]. In line with these findings, the GISTAR Pilot Study from Latvia also found that the low sensitivity of the PG panel may be a limiting factor regarding their use in population-based primary GC screening [56]. However, their high specificity could be helpful in triage [56].

In a study from Germany, serum PG screening was assessed in patients undergoing routine endoscopic evaluations (both upper and lower endoscopy from various indications) [57]. A PG(+) status was significantly correlated with gastric atrophic changes, with a relative risk for this condition of 12.2 (95% CI: 6.3–23.5). Moreover, patients with a high-risk GC profile according to the Operative Link of Gastritis Assessment (OLGA; stages III and IV) [58] could be identified by a serum PG assessment with a sensitivity of 75.0% and a specificity of 82.2%, respectively [57]. Therefore, the authors postulated that serological GC screening could be combined with a CRC-screening program, and individuals with a positive PG test should be offered an additional upper GI endoscopy in addition to screening colonoscopy [57].

Lastly, a multicenter study from the Netherlands and Norway showed that an assay of pepsinogen combined with a Gastrin-17 serological panel in patients with premalignant conditions of the stomach could be useful in stratifying those with higher or lower risk for malignant progression [59]. This study supports the role of serological markers in tailoring endoscopic surveillance programs for high-risk individuals in low-GC incidence areas.

In summary, a meta-analysis of 20 studies with a total of 4241 subjects has shown that a combination of pepsinogen, gastrin-17, and *H. pylori* antibody serological assays can reliably diagnose atrophic gastritis, with a sensitivity of 74.7% (95% CI, 62.0–84.3) and specificity of 95.6% (95% CI, 92.6–97.4) [60].

Another compelling marker that has been established constitutes the trefoil factor family proteins (TFF). Compared to PG testing alone, TFF3 was found to be more accurate with respect to GC diagnosis, with a sensitivity of 80.9% and a specificity of 81.0%. Combining TFF3 with PG I/II testing may provide an even more accurate non-invasive GC-screening modality [61]. However promising, TFF testing as a screening method has not yet entered clinical practice.

Taken together, a variety of tests addressing the different stages of the disease are increasingly available. A summary of postulated screening strategies for countries with different GC risk profiles is summarized in Table 2.

## 5. Future Perspectives of Gastric Cancer Screening

The recent progress in the knowledge of the molecular landscape of GC provides great potential for facilitating new screening modalities for early cancer diagnosis. For example, specific somatic mutations occurring in cancerous tissue can be potentially targeted and detected by polymerase chain reaction-based technologies [62]. Circulating tumor DNA (ctDNA) is an example of such a tumor biomarker that can be identified within a simple blood sample. In a landmark study by Bettegowda et al., which included 640 patients with four different cancer types, the specific ctDNA could be detected in 82% of solid tumors. Specifically with respect to gastroesophageal cancer, 57% of patients with localized disease and 100% with metastatic disease had detectable ctDNA in their blood samples [63].

More recently, a novel test, CancerSEEK, has been introduced to detect the cell-free DNA of different tumor types [64]. This test utilizes combined assays for both genetic alterations and protein biomarkers and provides the potential to identify the presence of cancer and localize its organ of origin. In a study incorporating 1005 patients with different types of potentially resectable solid tumors (stages I to III), the CancerSEEK test was positive in a median of 70% of cases. The sensitivities ranged from 69% to 98% for the detection of five cancer types (ovary, liver, stomach, pancreas, and esophagus), and for GC diagnosis alone, the specificity amounted to >70% (all stages combined) [64].

Other “liquid biopsy” tests targeting different circulating biomarkers, including exosomes, circulating RNAs, or cell-free DNA methylation profiles (the Galleri test [65]), are under investigation [66]. The prospect of early cancer detection (including GC) using an affordable, non-invasive blood test is appealing; however, further studies are needed to establish the performance of these tests in the general population while considering potential challenges such as the handling of false-positive results.

## 6. Conclusions

Given the significant variations in both GC incidence (from 0.75 in Mozambique to 32.5 per 100,000 population in Mongolia) and income per capita around the globe, screening strategies for GC should be adjusted to the actual conditions in each region. While several proven tools exist for accurate GC diagnosis, a universal modality for GC populational screening remains elusive.

An endoscopy in qualified hands is an excellent tool for diagnosing GC. However, although accurate in diagnosing early GC and precancerous conditions (especially in recent developments, including the widespread use of magnifying endoscopy and image-enhancement techniques [67]), an endoscopy-based screening program for GC results in a sizeable socioeconomic burden and the consumption of medical resources. Such a strategy is only cost-effective in high-GC-incidence regions. Additionally, it focuses solely on secondary prevention (early detection) rather than the primary prevention of cancer.

The “test-and-treat” strategy for *H. pylori* infection has already been established in several countries and has contributed to decreasing GC incidence in some regions. In western countries, this strategy is now most commonly applied to patients with dyspeptic symptoms and has been proven to reduce the costs of dyspepsia work-ups in general healthcare [68]. With this in mind, the use of a more general testing (including also asymptomatic individuals) scheme on a populational level may constitute a feasible strategy within healthcare systems. However, this concept contains inherent challenges, such as increasing antibiotic resistance and possible adverse effects, which require thorough consideration. Additionally, emerging biomarker-based strategies, including PG assays, may help identify at-risk individuals and constitute a more patient-friendly alternative to replace endoscopy as a primary diagnostic modality.

It is worth emphasizing that there is a general lack of interventional healthcare studies within the field of GC screening, especially in western countries. Nevertheless, this critical medical and social demand could surely be the prelude to an exciting era of research, thereby furthering the fight against a disease that remains a persistent threat to a large subset of the population throughout the globe.

## Figures and Tables

**Figure 1 cancers-15-00664-f001:**
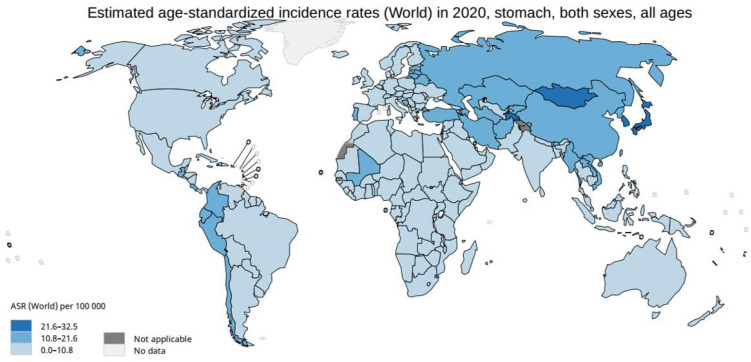
Age-standardized incidence rates (ASRs) of gastric cancer worldwide.

**Table 1 cancers-15-00664-t001:** Overview of screening programs in high-risk countries.

	Japan [18,19]	South Korea [20,21]	China—UGCED Program	China—CanSPUC Program
Year of implementation:	1983 (last updated in 2018)	1999 (last updated in 2015)	2008 (last updated in 2020)	2012
Coverage:	Nationwide	Nationwide	High-risk rural areas	High-risk urban areas
Screening test:	UGIS or EGD	UGIS or EGD (EGD recommended)	EGD	Questionnaire + HP and EGD
Target age for screening:	≥50 years (no upper age limit)	≥40 (no upper age limit)	40–69 years	40–69 years
Screening interval:	Every 2–3 years	Every 2 years	Individuals diagnosed with severe CAG/IM and LGD: repeated endoscopy within 3 years	CAG/IM/ gastric polyps: repeated endoscopy within 6–12 months; LGD: repeated endoscopy within 3–6 months
Compliance:	48.0% (2019)	45.4% (2011) [22]	NA.	NA

CanSPUC, Cancer-screening program in Urban China; EGD, esophagogastroduodenoscopy; HP, *Helicobacter Pylori* antibody testing; UGCED, Upper Gastrointestinal Cancer Early Detection; UGIS, upper gastrointestinal series.

**Table 2 cancers-15-00664-t002:** Summary of established and postulated screening strategies for gastric cancer in different risk areas.

Low-Risk Areas (ASR < 10 per 100,000)	Intermediate-Risk Areas (ASR ≥ 10 and <20 per 100,000)	High-Risk Areas (ASR ≥ 20 per 100,000)
Targeted GC screening for at-risk individuals (postulated and potentially cost-effective)		Primary GC screening (established and cost-effective)
*H. pylori* “screen-and-treat” method for at-risk individuals (e.g., family history of GC, precancerous gastric lesions)		*H. pylori* “screen-and-treat” general population
Serological testing (serum pepsinogen) in high-risk individuals (e.g., smoking men over 50 years of age)	Upper GI endoscopy in FOBT-positive CRC screening individuals	Primary-imaging screening (upper GI endoscopy/gastrography) for individuals ≥40 (50) years old.
Stool antigen *H. pylori* testing combined with a FOBT-based CRC screening program		
Serological testing (serum pepsinogen) coupled with CRC-screening program		

ASR; age-standardized ratio, CRC; colorectal cancer, GC; gastric cancer, FOBT; a fecal occult blood test.

## Data Availability

Not applicable.

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
