# Peer review of "A Global Perspective on Gastric Cancer Screening: Which Concepts Are Feasible, and When?"

_cancers, 2023, doi:10.3390/cancers15030664_

Round 1
Reviewer 1 Report
The authors present an overview about gastric cancer screening. The review is well written and comprehensively covers several screening approaches for the screening of gastric cancer.
I have, however, some remarks
- Introduction: the authors should use the current WHO classification (2019) for the description of gastric cancer subtypes and also briefly mention EBV associated and MSI gastric cancer, and then additionally refer to the Laurén classification. There are some well written reviews about current classification systems of gastric cancer (e.g. Smyth et al., Lancet 2020) which could serve as reference.
- Some words about the aims and the focus of the review before paragraph 1.1 would enhance the readability of the paper
- Serological biomarkers, lines 244ff. It seems to be confusing at first glance to have PG screening in patients undergoing colonoscopy. In ref 51, half of the patients also had gastroscopy which enabled the authors to correlate the serologic findings with the presence of detectable risk factors. Although this issue is explained later, I would suggest to mention it earlier.
- The authors do not mention any studies about the efficacy of the presented screening strategies. Is there any evidence for a benefit both regarding health care and economics and if not, then it should be discussed.
- The formatting of the references does not fit to the layout of the main manuscript. Some references have capital letters only which may be a formatting error but should be corrected before final submission.
Reviewer 2 Report
This is a comprehensive review of the literature concerning population screening for early detection of gastric cancer.
Authors performed a practical stratification of screening results and advantages according to the different incidence of the disease in different countries. Furthermore, they focused on biochemical detection of the major GC promoter, Helicobacter Pylori.
I suggest the acceptance of this manuscript
Author Response
Thank you for appreciating our work and agreeing to publish our article.
Reviewer 3 Report
This paper is a nareative review upon screening of gastric cancer, non cardial in different regions on the globe especially in those with high risk. The method for literature research is adequate, important and significant articles are retrieved and examined. The autors have summarized the up to date literature and this is what offers to the reader. Overall I think it is a good work, although it doesn’t offer an answer to the problem it raises and opens some questions and ideas to be approached in the future research.
Author Response
Thank you for appreciating our work, surely the article leaves several questions unanswered, but hopefully, it provides a comprehensive and informative review of the existing knowledge on the topic of GC screening. Many thanks for agreeing to publish our work.
Reviewer 4 Report
Gastric cancer has remained a huge disease burden worldwide. Compared with the limited survival of advanced GC, the early GC has an excellent prognosis, which emphasized the importance of cancer screening. In this reviewer, the author reviewed the existing screening strategies for GC in high-risk regions and discuss potentially applicable modalities in countries with low-to-intermediate incidence. Moreover, the author also discussed about the test of H. pylori, the class I carcinogen of GC. References for this Review article are comprehensive, and the article was well-organized. While there are still some concerns about this review.
1. The author reviewed several traditional screening methods of GC, while some novel means of cancer screening, including circulating tumor cells, exosomes, or some others have been gradually applied. The prospection of these methods in GC screening could also be discussed.
2. The author focused mainly on the non-cardia GC. However, recent years, the incidence rate of the adenocarcinoma of esophagogastric junction (AEG) has been rising around the world. Are there any screening strategies specific for this type of gastric cancer?
Round 2
Reviewer 4 Report
After revision, the author added a separate paragraph about the future perspectives of gastric cancer screening which enriched the content of this review. The article is well-organized and could be accepted in present form.